# VISION LANGUAGE MODELS INHERIT HUMAN COLOR PERCEPTION

## ABSTRACT

Vision language models (VLMs) receive raw sRGB pixel values and could, in principle, discriminate colors at machine precision. But do they? And if not, what determines their perceptual thresholds? We use psychophysics-inspired experiments to characterize the color discrimination boundary of two VLMs (Gemini 3 Flash and Qwen3-VL-8B-Instruct) and ask which color-distance metric best explains it. Across three tasks (odd-one-out, same/different, triplet matching), two models, and both 2D chromaticity and full 3D CIELAB color spaces (totaling over 68,000 trials), CIE $\Delta E_{00}$ (a metric engineered to match human perception) consistently outperforms all input-space metrics, including sRGB L2, linear RGB L2, and CIE XYZ L2. Residual analysis confirms that $\Delta E_{00}$ is a sufficient statistic for VLM sensitivity in the chromaticity plane, though systematic axis-dependent deviations emerge when lightness varies. Layerwise probing of Qwen3-VL-8B-Instruct reveals that patch embeddings strongly prefer sRGB ($R^2{=}0.97$) over $\Delta E_{00}$ ($R^2{=}0.46$), indicating that perceptual structure is not built into the input projection but emerges downstream in the network. Overall, we demonstrate that VLMs, through large-scale training, have inherited the perceptual color space of the humans involved in data generation.

## 1 INTRODUCTION

What are the sensory systems of a neural network? In this paper, we attempt a fusion of psychophysics and deep learning: we carefully test the color sensitivity of vision language models and ask whether it resembles human perception.

Vision language models (VLMs) combine pretrained vision encoders with large language model backbones (Zhang et al., 2024), enabling them to reason about images through natural language. These models receive raw sRGB pixel values as input (Anderson et al., 1996), a coordinate system designed for display hardware, not human perception. In principle, they could discriminate colors at machine precision. But do they? And if not, do their perceptual thresholds follow sRGB geometry, or something else entirely?

We hypothesize that through training on massive human-generated data, VLMs have inherited the sensory biases of their data's curators: humans. Color choices in photographs, labels, and visual designs all reflect human perceptual categories. If VLM sensitivity contours match human perceptual contours (MacAdam, 1942; 1943) (as captured by the CIE $\Delta E_{00}$ metric (Sharma et al., 2005; Fairchild, 2013)), this would suggest the model has absorbed human sensory structure from data. The alternative hypothesis is simpler: perhaps VLMs merely operate as distance calculators in their native input space (sRGB), with no perceptual inheritance.

We test these hypotheses using psychophysics methodology (controlled stimuli, forced-choice tasks, and psychometric curve fitting) across two VLMs, three task formats, and over 68,000 trials. Our contributions are:

1. **VLM sensitivity follows human perceptual metrics.** We show that VLM color sensitivity is better explained by the human perceptual metric $\Delta E_{00}$ than by sRGB or any other input-space metric. The effect replicates across models, tasks, and both 2D chromaticity and 3D CIELAB color spaces.

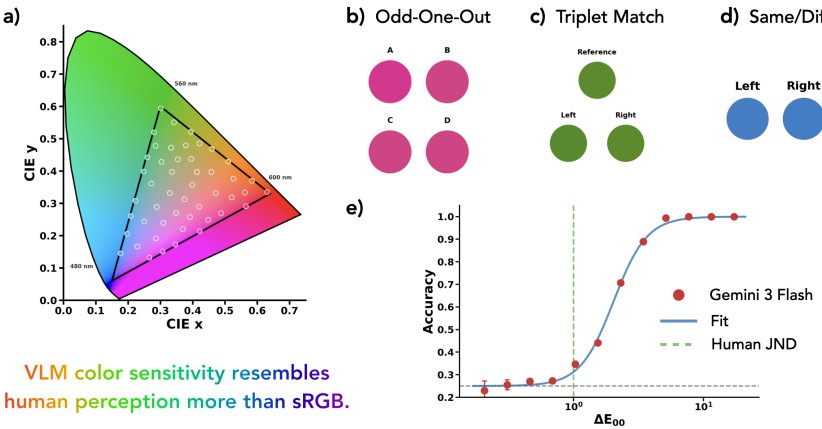

Figure 1: **Overview.** (a) CIE xy chromaticity diagram showing the spectral locus (colored horse-shoe), sRGB gamut (triangle), and our 48 experimental base colors (black dots). (b–d) Three forced-choice tasks probe VLM color discrimination: odd-one-out (4AFC), same/different (2AFC), and triplet matching (2AFC). (e) Psychometric curve for Gemini 3 Flash on odd-one-out: accuracy rises from chance (25%) to ceiling as $\Delta E_{00}$ increases. The dashed green line marks the human just-noticeable difference (JND $\approx 1$ $\Delta E_{00}$); at this threshold, the VLM remains at chance.

2. **Perceptual structure emerges in deeper layers.** We decompose where this perceptual structure originates: layerwise probing reveals it is absent at the input projection and emerges progressively through the vision transformer. Comparing eight vision encoders, we find that training objective predicts perceptual alignment: image-text contrastive models achieve highest $\Delta E_{00}$ $R^2$, while self-supervised pixel reconstruction (MAE) achieves lowest.

## 2 METHODS

We evaluate two VLMs (Gemini 3 Flash and Qwen3-VL-8B-Instruct) on forced-choice color discrimination tasks using synthetic solid-color stimuli. We sample color pairs in both 2D chromaticity (1,677 pairs at fixed luminance) and 3D CIELAB space (5,160 pairs), presented as odd-one-out (4AFC), same/different (2AFC), and triplet matching (2AFC) tasks. For each trial, we fit psychometric functions predicting accuracy from color distance under competing metrics: sRGB L2, linear RGB L2, XYZ L2, $\Delta E_{76}$, and $\Delta E_{00}$. We compare models via log-likelihood and BIC. For representation analysis, we extract layerwise activations from Qwen3-VL-8B-Instruct and measure $R^2$ between embedding distances and each color metric. See App. B for full details.

## 3 RESULTS

### 3.1 $\Delta E_{00}$ BEST EXPLAINS VLM COLOR DISCRIMINATION

Table 1 presents the 7-model comparison across all evaluated datasets. $\Delta E_{00}$ achieves the highest log-likelihood in every condition except one (Gemini 2D, where a 5-parameter per-channel power-law model marginally outperforms the 2-parameter $\Delta E_{00}$ by $\Delta LL=22$ nats but loses on BIC in the 3D dataset). Bootstrap resampling confirms these rankings are robust (100% agreement across 1,000 resamples).

However, the ordering of fixed metrics does *not* follow the colorimetric transform chain. In particular, sRGB L2 outperforms both linear RGB L2 and XYZ L2. Undoing the sRGB gamma encoding (which moves the representation *toward* physical light intensity) *hurts* predictive power ($\Delta LL= -172$ nats for Gemini 2D). The largest single improvement occurs at the XYZ→CIELAB step ($\Delta LL= +489$ nats), where the perceptual cube-root compression enters. The further refinements of $\Delta E_{00}$ add another +94 nats over plain CIELAB.

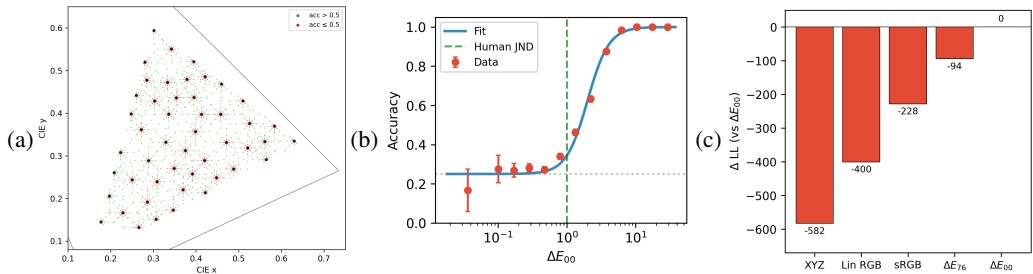

Figure 2: **Behavioral results (Gemini 2D).** (a) CIE xy chromaticity diagram showing our 48 base points (black) and their neighbors colored by accuracy (green: >50%, red: ≤50%). (b) Psychometric curve: accuracy vs $\Delta E_{00}$ with logistic fit; dashed green line marks human JND ($\Delta E_{00}{=}1$). (c) Log-likelihood difference from $\Delta E_{00}$ baseline for competing metrics. $\Delta E_{00}$ outperforms all input-space metrics (sRGB by 228 nats, XYZ by 582 nats).

Table 1: Log-likelihood of psychometric fits under competing color-distance models. Higher (less negative) is better. The 2D experiments sample 1,677 pairs (6,706–6,708 trials); the 3D experiments sample 5,160 pairs (20,639–20,640 trials). Best fixed-metric LL in **bold**.

| Metric | 2D (chromaticity) | | 3D (CIELAB) | |
| --- | --- | --- | --- | --- |
| | Gemini | Qwen3 | Gemini | Qwen3 |
| XYZ L2 | $-3640$ | $-3905$ | $-11280$ | $-11939$ |
| Linear RGB L2 | $-3458$ | $-3702$ | $-10781$ | $-11424$ |
| sRGB L2 | $-3285$ | $-3727$ | $-9157$ | $-10463$ |
| $\Delta E_{76}$ | $-3152$ | $-3487$ | $-9076$ | $-10174$ |
| $\Delta E_{00}$ | $\mathbf{-3058}$ | $\mathbf{-3390}$ | $\mathbf{-8816}$ | $\mathbf{-9909}$ |
| sRGB$^{\gamma}$ (shared) | $-3125$ | $-3503$ | $-9140$ | $-10379$ |
| sRGB$^{\gamma}$ (per-ch) | $-3035$ | $-3462$ | $-9126$ | $-10334$ |

The same qualitative ordering replicates across both 2AFC tasks (same/different and triplet matching), with $\Delta E_{00}$ LL of $-1706$ and $-1628$ respectively for Gemini, compared to sRGB LL of $-1730$ and $-1677$.

## 3.2 $\Delta E_{00}$ IS SUFFICIENT IN 2D BUT NOT IN 3D

In the 2D chromaticity plane, the 2-parameter $\Delta E_{00}$ model has the lowest BIC across all conditions (Table 2). Adding 47 base-point or 7 direction intercepts produces statistically significant LRT $p$-values ($p < 0.001$) but *increases* BIC, indicating that the effects are real but too small to justify the additional parameters. The $\Delta E_{00}$ model captures VLM sensitivity across the chromaticity plane without systematic residuals.

In 3D CIELAB space, the picture changes. Adding a single L-vs-chromatic intercept, distinguishing lightness ($L^*$) directions from chromatic ($a^*$, $b^*$) directions, reduces BIC by 27 nats (Gemini) and 113 nats (Qwen3). Both models find chromatic axis $a^*$ easiest and $b^*$ hardest, with accuracy of 63.9% vs. 57.1% (Gemini) and 51.0% vs. 46.6% (Qwen3). $\Delta E_{00}$ does not fully equalize VLM sensitivity across the L*, a*, b* axes, suggesting residual structure beyond what the standard perceptual metric captures.

## 3.3 PERCEPTUAL STRUCTURE EMERGES IN DEEPER LAYERS

Figure 3 summarizes the representational analysis of Qwen3-VL-8B-Instruct. At the patch embedding layer, the $R^2$ between representation distance and sRGB L2 is 0.97, while $R^2$ with $\Delta E_{00}$ is only 0.46 (Figure 3a,b). The input projection preserves sRGB geometry almost perfectly; perceptual structure is not at the gate.

Table 2: Residual analysis: BIC of $\Delta E_{00}$ models with and without spatial/directional intercepts. Lower is better. In 2D, $\Delta E_{00}$-only wins. In 3D, adding an L-vs-ab direction intercept (1 extra parameter) wins for both models.

| Model | 2D Odd-one-out | | 3D Odd-one-out | |
|---|---|---|---|---|
| | Gemini | Qwen3 | Gemini | Qwen3 |
| $\Delta E_{00}$ only (2p) | **6133** | **6798** | 17651 | 19838 |
| + direction type (3–4p) | 6138 | 6794 | **17624** | **19725** |
| + smooth (8–11p) | 6160 | 6810 | 17650 | 19757 |
| + base point (49p) | 6325 | 6978 | 17788 | 20008 |

Across the vision transformer layers, sRGB $R^2$ drops steadily while $\Delta E_{00}$ $R^2$ rises, crossing over at layer 10 (Figure 3c). We observe the same pattern across eight standalone vision encoders with varying training objectives (Figure 3d): all begin with high sRGB $R^2$ ($>0.90$) at patch embedding, but final-layer $\Delta E_{00}$ $R^2$ depends on training objective. Image-text contrastive models (CLIP, SigLIP) and supervised classification (ViT-IN21k) achieve highest perceptual alignment ($R^2$ 0.61–0.76), while self-supervised reconstruction (MAE) achieves lowest ($R^2 = 0.39$).

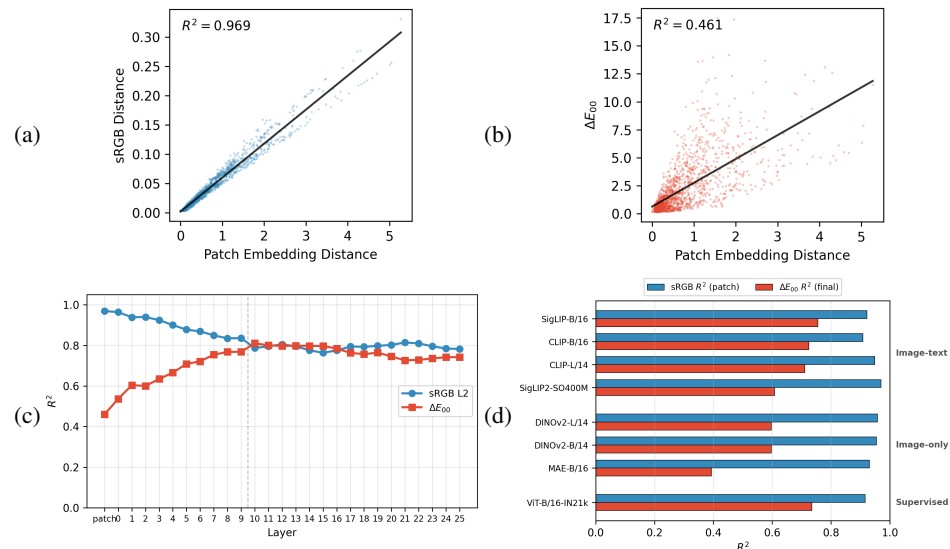

Figure 3: **Embedding analysis.** (a,b) Patch embedding distance correlates strongly with sRGB ($R^2$=0.97) but weakly with $\Delta E_{00}$ ($R^2$=0.46). (c) Across ViT layers, sRGB $R^2$ drops while $\Delta E_{00}$ $R^2$ rises; crossover occurs at layer 10. (d) Across eight vision encoders, all preserve sRGB at patch embedding (blue); final-layer $\Delta E_{00}$ $R^2$ (red) varies by training objective.

## 4 CONCLUSION

We find that VLM color discrimination is best explained by $\Delta E_{00}$, a metric designed to match human perception, rather than by any input-space metric. This perceptual structure is absent at the patch embedding (where sRGB $R^2 = 0.97$ dominates) and emerges in deeper layers, with the degree of alignment depending on training objective: image-text models achieve strongest $\Delta E_{00}$ alignment, while self-supervised reconstruction (MAE) achieves weakest. Our results demonstrate that through large-scale training, neural networks can inherit perceptual structures from the humans involved in data curation.

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

## A    RELATED WORK

**Color perception in VLMs.**    Recent work has begun to probe VLM color understanding. ColorSwap (Burapacheep et al., 2024) tests color-word binding with swapped color terms, ColorFoil (Samin et al., 2025) investigates color blindness patterns in VLMs, and ColorBench (Liang et al., 2025) provides a comprehensive benchmark evaluating VLMs on color perception, reasoning, and robustness, finding that current models struggle with fine-grained color tasks. Our work differs in focus: rather than benchmarking task performance, we ask *which metric* best predicts VLM color discrimination and *where* the perceptual structure originates in the network.

**Psychophysics and machine perception.**    Psychophysics (Gescheider, 2013), founded by Weber and Fechner (Weber, 1834; Fechner, 1860), later refined by Stevens (Stevens, 1957), and formalized through signal detection theory (Green & Swets, 1966), studies the quantitative relationship between physical stimuli and perception. Classical color science established that human color discrimination is non-uniform across color space (MacAdam, 1942; 1943), motivating perceptually uniform metrics like CIELAB and $\Delta E_{00}$ (Sharma et al., 2005; Fairchild, 2013). We adapt psychophysical methodology (forced-choice tasks, psychometric curve fitting (Wichmann & Hill, 2001a;b)) to characterize VLM perception, enabling direct comparison between machine and human sensitivity profiles.

**Learned representations and human perception.**    Whether deep networks are adequate behavioral models of human perception remains debated (Wichmann & Geirhos, 2023; Funke et al., 2021). Prior work has identified both alignments and divergences, such as texture bias in CNNs (Geirhos et al., 2019). Our layerwise analysis extends this to color, showing that perceptual structure emerges progressively through the vision transformer (Dosovitskiy et al., 2021) rather than being imposed by the input encoding.

## B    METHODS DETAILS

### B.1    STIMULUS DESIGN

**2D chromaticity stimuli.**    We sample 48 base points in the CIE 1931 xy chromaticity plane (at fixed luminance $Y = 0.2$) using furthest-point selection for spatial coverage within the sRGB gamut. For each base point, we generate neighbor colors at 5 radii in xy-space ($r \in \{0.002, 0.004, 0.008, 0.016, 0.032\}$) along 8 uniformly spaced directions ($0°$–$315°$), yielding 1,677 valid color pairs after gamut filtering.

**3D CIELAB stimuli.**    To test the full perceptual metric including lightness, we sample 48 base points in CIELAB space via rejection sampling (ensuring sRGB gamut membership) and generate neighbors at 5 radii ($r \in \{1, 2, 4, 8, 16\}$ $\Delta E_{76}$ units) along all 26 directions in $\{-1, 0, 1\}^3$, yielding 5,160 valid pairs.

**Tasks.**    Each color pair is rendered as a visual forced-choice task:

- **Odd-one-out (4AFC):** Four circles on a $2 \times 2$ grid; three share the base color, one has the neighbor color. All 4 positions are tested per pair. Chance = 25%.
- **Same/different (2AFC):** Two circles; the pair is either identical (same) or different. Chance = 50%.
- **Triplet matching (2AFC):** A reference circle plus two candidates (one matching, one different). Chance = 50%.

### B.2    VLM EVALUATION

We evaluate two VLMs: **Gemini 3 Flash** (via the Gemini API) and **Qwen3-VL-8B-Instruct** (run locally). Each model receives the stimulus image and a text prompt requesting the forced-choice answer. We use low-temperature generation ($T = 0$) with a single retry at $T = 0.6$ for unparseable responses. Parse rates exceed 99.9% across all conditions.

## B.3 COLOR-DISTANCE MODELS

We fit psychometric functions (logistic curves) predicting accuracy from log-transformed color distance under seven competing metrics:

1. **sRGB L2**: Euclidean distance in the model's native input space.
2. **Linear RGB L2**: After undoing sRGB gamma ($\gamma \approx 2.2$).
3. **CIE XYZ L2**: After the linear $3 \times 3$ transform to CIE 1931 tristimulus.
4. $\Delta E_{76}$: Euclidean distance in CIELAB (cube-root compression of XYZ).
5. $\Delta E_{00}$: The full CIE 2000 perceptual metric with hue rotation, chroma-dependent weighting, and lightness corrections.
6. **sRGB$^{\gamma}$ (shared)**: $\|(\text{sRGB})^{\gamma}\|_2$ with a single learned $\gamma$ (3 free parameters).
7. **sRGB$^{\gamma}$ (per-channel)**: Separate $\gamma_R, \gamma_G, \gamma_B$ (5 free parameters).

Models 1–5 each have 2 free parameters (slope $k$ and midpoint $x_0$). We compare all models using log-likelihood (LL) with 1,000 bootstrap resamples and Bayesian Information Criterion (BIC).

## B.4 RESIDUAL ANALYSIS

To test whether $\Delta E_{00}$ is a *sufficient* predictor, we augment the $\Delta E_{00}$ logistic model with additional intercept terms: (a) per-base-point intercepts (48 levels), (b) per-direction intercepts (8 levels in 2D, or L/a/b axis-type in 3D), and (c) smooth spatial terms. We compare via BIC and likelihood ratio tests (LRT).

## B.5 REPRESENTATION ANALYSIS

We extract internal representations from Qwen3-VL-8B-Instruct (the only model with open weights) by passing solid-color images through the network and recording activations at every layer. For each of the 1,677 color pairs, we compute L2 distance in representation space at each layer and measure its $R^2$ correlation with each of the five fixed color metrics. This traces where the representational geometry transitions from sRGB-like to $\Delta E_{00}$-like.

