# OpenReview forum: "Vision Language Models Inherit Human Color Perception"
_ICLR.cc/2026/Workshop/Sci4DL — Sci4DL 2026_

### Official Review · Reviewer_uey4 · 2026-02-20

**Fit:** 3
**Significance:** 2
**Confidence:** 2

**Summary:**

This paper explores whether color discrimination of VLMs aligns with machine-level perception (sRGC/RGB color spaces) or human perception. Using a psychophysics-inspired methodology, the authors test two models across three different task formats: odd-one-out, same/different, and triplet matching. With 68000 trials in 2D and 3D CIELAB color spaces, results demostrate that CIE$\Delta E_{00}$ metric, specifically designed to model human color perception, outperforms all input-space metrics in predicting VLM sensitivity. Layerwise analysis of different image embeddings ViT models reveals that initial patch embeddings are highly correlated with raw sRGB values, while human-like perceptual structures emerge in the deeper layers of the network.

**Strengths:**

There are multiple strengths for this paper:
- It addresses a fundamental and novel question in both AI interpretability and AI-human alignment: Do the models see the world through raw digital inputs, or do they have a human-like perception?
- The use of classic psychophysics techniques and different tasks provides a very robust framework.
- With the layerwise experiment, the authors show where in the network hierarchy these perceptual representations emerge.

**Suggestions:**

Some suggestions, things that should be clarified, and recommendations for future work for the authors:

1. Technical Clarification on Psychometric Curves

In Section B.2, the authors state they used low-temperature generation ($T=0$) with a single retry at $T=0.6$ for unparseable responses. If each image-pair configuration was tested only once, it is unclear how the smooth psychometric functions (e.g., Figure 1) were derived, as $T=0$ should produce binary (0 or 1) results for a given input. Even if the accuracies are aggregated across the different possible spatial configurations (as mentioned in B.1) for each experiment, this only gives 4 runs for the 4AFC, which is still very low for a smooth curve. Therefore, does it mean that you are aggregating across different experiments, different experiment configurations and more importantly, around different base colors? If so, how do the authors account for the potential variance in sensitivity across different regions of the color space?

2. Task-Specific Variance and Aggregation

The paper currently presents summarized results, but three different experiment configurations with with different chance levels are used. It would be valuable to see the psychometric functions plotted separately for each task. This would clarify if certain task formats (like triplet matching vs. same/different) make the model more or less sensitive to color differences.

3. Reproducibility and Prompting

To ensure the results are not influenced by prompt engineering, the authors should include the exact system and user prompts used for each of the three tasks in the appendix.

4. Ambiguity in Layerwise Analysis

The paper mentions extracting internal representations of Qwen3-VL-8B-Instruct but then it talks about the Vision encoder. Therefore, are the layerwise results about the ViT (Vision encoder) or of the LLM? Clarifying this is essential to understanding whether it happens during visual encoding or linguistic processing.

Does the "cross-over" point (Figure 3c) occur at a consistent relative depth for all the models? Or does it happen at layer 9 for all the models independently of their number of layers?

5. Intra-family and Scaling Analysis

While appropriate for a workshop, the paper would be significantly strengthened by including an analysis of how model size affects this inheritance. For example, investigating whether smaller models (e.g., Qwen-VL-2B) exhibit the same human-like biases as larger ones would help determine if this is a scaling law of "perceptual alignment."

---

### Official Review · Reviewer_ffH3 · 2026-02-24

**Fit:** 3
**Significance:** 2
**Confidence:** 2

**Summary:**

The paper "Vision Language Models Inherit Human Color Perception" examines the color discrimination capabilities of VLMs, and asks in particular whether this color sensitivity aligns more with "pure" machine color space---as could be expected given that the models are fed images in sRGB encoding---or with human perceptual color space. The surprising answer is that the investigated models exhibit color sensitivity that best matches human perceptual color space. The conclusion is that the models inherited human color sensitivity through their human-generated training data. Moreover, this sensitivity does not manifest at the gate, where as expected conformity with sRGB dominates, but gradually appears in deeper layers of the model.

**Strengths:**

This work is a great example of how tools from psychology can be applied to computer models to gain a better understanding of their inner workings, and to assess their correspondence to human behavior. The experimental setup is well thought out and elegantly executed and the obtained results provide some nutritious food for thought!

**Suggestions:**

The result that the investigated models align so well with human color perception is quite surprising. You state that (quoting the abstract): "Overall, we demonstrate that VLMs, through large-scale training, have inherited the perceptual color space of the humans involved in data generation." How exactly did the model inherit human color perception through the training data, do you think? Do you have access to the original training data? My point is this: the (large-scale) data used to train these models is undoubtedly very diverse. To inherit this specific behavior, I would expect the models to have been trained using data similar to the experiment you used, but is this the case? I.e., did the model truly inherit this human perceptual color sensitivity, or did it just happen to be the case that the training data contained a lot of similar psychophysics visual color perception questions, and so the model simply *appears* to showcase this sensitivity in this particular context, as it has learned to emulate *human responses* to these types of questions?

Have you explored alternative prompts where you, e.g., explicitly instructed the model to "answer like a human would" or alternatively to "be as precise as you can be"? This might help answer my previous question.

Despite the great experimental approach and the significance of the obtained results, the manuscript could certainly be improved in multiple ways. Many small inconsistencies make the writing come over as rather sloppy. A non exhaustive list with suggestions and comments follows:
- Several acronyms are never or unsatisfactoraly explained: AFC (never explained), BIC (only explained in appendix), JND (only explained in caption of Figure 1, should be explained in main text), LRT (never explained). The "LL" in line 100 should have been explained in line 99: "log-likelihood" --> "log-likelihood (LL)".
- Figures 1 and 2 are never referenced in the main text, while they should be.
- What is the added value of Figure 2.b over Figure 1.e?
- Avoid using the word "carefully" in line 32; all papers are expected to report on carefully performed experiments!
- Line 85/86: no references are provided to the Gemini 3 and Qwen3 models.
- Line 92: "App." --> "Appendix" (i.e., please write in full)
- Some of the mentioned numerical values in the main text don't agree with the values in the tables: line 100 mentions a 22 nats difference that is 23 in Table 1; line 106 mentions -172 that reads as -173 in Table 1; line 107 mentions +489 that should in fact be +499 unless I'm mistaken; it is unclear to me to what the +94 nats difference in line 107 refers exactly (I can not find this back in Table 1).
- Line 106 mentions "XYS $\rightarrow$ CIELAB"; shouldn't this be "XYS $\rightarrow$ Linear RGB L2 for CIELAB"?
- Additionaly, that +489 difference is quoted as being the largest difference, however the "XYS $\rightarrow$ Linear RGB L2 for Qwen3" difference is actually +515.
- It is not clear for what task(s) Figure 2 reports results. The mentioned 228 difference reads as 227 in Table 1.
- Line 209/210: "(where sRGB $R^2$ = 0.97 dominates)" reads a bit strange; I would suggest rephrasing to something like "(where sRGB dominates with $R^2 = 0.97$)".
- Line 322: "Each model receives the stimulus image and a text prompt requesting the forced-choice answer." Although it can arguably be derived from the context to what "the stimulus image" and "the forced-choice answer" refers, I would suggest to rephrase this line to more explicitly state to what image and what answer exactly you refer.

---

### Official Review · Reviewer_hHsg · 2026-02-27

**Fit:** 2
**Significance:** 1
**Confidence:** 2

**Summary:**

This paper uses psychophysics-inspired experiments to investigate whether Vision Language Models (VLMs) perceive color like humans or according to their raw digital input space (sRGB). Through over 68,000 trials across two models (Gemini 3 Flash and Qwen3-VL-8B-Instruct), the authors demonstrate that VLM color discrimination is best explained by $\Delta E_{00}$, a metric specifically engineered to match human perception. Their findings suggest that these models "inherit" human sensory biases through large-scale training on human-curated data, with this perceptual structure emerging in the deeper layers of the network rather than the initial input stage.

**Strengths:**

1. The study employs established psychophysical techniques, including forced-choice tasks (odd-one-out, triplet matching, same/different) and psychometric curve fitting, providing a robust framework for comparing machine and human perception.
2. The findings are backed by a massive dataset of over 68,000 trials across both 2D chromaticity and full 3D CIELAB color spaces, ensuring statistical significance and cross-task consistency.

**Suggestions:**

Weakness:

1. While the study tests two prominent VLMs, both are from relatively recent architectures; testing older models or a wider variety of backbones (e.g., non-Transformer architectures) would strengthen the "inheritance" claim.

2. he evaluation is restricted to solid-color synthetic stimuli. While necessary for psychophysics, it remains unclear how these "inherited" perceptual thresholds translate to more complex, real-world images with textures, lighting gradients, and shadows.

3. The residual analysis reveals that the $\Delta E_{00}$ metric does not fully explain VLM sensitivity when lightness $(L^*)$ varies, suggesting the models have residual structural biases that current human-centric metrics don't perfectly capture.

---

### Meta-Review · Area_Chair_hMJL · 2026-02-28

**Recommendation:** Accept

**Metareview:**

This paper compares color perception in humans and VLMs using a methodology inspired from psychophysics. All reviewers noted the originality and quality of the experiments. I recommend acceptance.

---

### Decision · Program_Chairs · 2026-03-02

Accept